# PINK1 Phosphorylates Drp1^S616^ to Improve Mitochondrial Fission and Inhibit the Progression of Hypertension-Induced HFpEF

**DOI:** 10.3390/ijms231911934

**Published:** 2022-10-08

**Authors:** Jian Shou, Yunlong Huo

**Affiliations:** 1Institute of Mechanobiology and Medical Engineering, School of Life Sciences and Biotechnology, Shanghai Jiao Tong University, Shanghai 200240, China; 2PKU-HKUST Shenzhen-Hong Kong Institution, Shenzhen 518000, China

**Keywords:** Drp1, HFpEF, mitochondrial fission, PINK1

## Abstract

(1) Background: Heart failure with preserved ejection fraction (HFpEF) is a major subtype of HF with no effective treatments. Mitochondrial dysfunctions relevant to the imbalance of fusion and fission occur in HFpEF. Drp1 is a key protein regulating mitochondrial fission, and PINK1 is the upstream activator of Drp1, but their relationship with HF has not been clarified. The aim of the study is to investigate molecular mechanisms of mitochondrial dysfunctions in animals with hypertension-induced HFpEF. (2) Methods and Results: The hypertension-induced HFpEF model was established by feeding Dahl/SS rats with high salt, showing risk factors such as hypertension, mitochondrial dysfunctions, and so on. Physiological and biological measurements showed a decrease in the expression of mitochondrial function-related genes, ATP production, and mitochondrial fission index. PINK1 knockout in H9C2 cardiomyocytes showed similar effects. Moreover, PINK1 myocardium-specific overexpression activated Drp1^S616^ phosphorylation and enhanced mitochondrial fission to slow the progression of hypertension-induced HFpEF. (3) Conclusions: PINK1 could phosphorylate Drp1S616 to improve mitochondrial fission and relieve mitochondrial dysfunctions, which highlights potential treatments of HFpEF.

## 1. Introduction

Heart failure (HF) is a complex syndrome with systolic or diastolic dysfunctions [1]. This reduces heart pumping capacity and leads to problems in satisfying metabolic requirements of human organs and tissues. Heart failure with reduced ejection fraction (HFrEF) and heart failure with preserved ejection fraction (HFpEF) are two main subtypes of HF [2,3]. The prevalence of HFpEF is increasing in China [4]. Risk factors of HFpEF include diabetes, obesity, and hypertension, showing diastolic dysfunctions, myocardial hypertrophy, interstitial fibrosis and other phenotypes [3,5,6,7]. Substantial mitochondria in myocytes are of importance in regulating HFpEF [8,9]. Risk factors for HFpEF could induce mitochondrial dysfunctions [10]. Hence, mitochondrial-targeted therapy may be effective for HFpEF.

The balance of fusion and fission maintains mitochondrial functions [11,12]. The fusion depends on mitofusin 1/2 (Mfn1/2) and optical atrophy 1 (OPA1) to enhance the respiratory capacity of mitochondria [13]. The fission is mainly completed by dynamic related protein 1 (Drp1) removing damaged parts [14]. Drp1 is transferred to the outer membrane of mitochondria [15] and binds to mitochondrial fission factor (MFF) and fission protein 1 (Fis1) to promote mitochondrial fission [16,17]. PTEN-induced putative kinase 1 (PINK1), a serine/threonine protein kinase in mitochondria and cytoplasm [18,19,20,21,22], could directly phosphorylate Drp1^S616^ to regulate mitochondrial fission [23]. There is, however, a lack of studies to show how PINK1 phosphorylates Drp1^S616^ to regulate mitochondria during the occurrence and development of HFpEF.

The aim of this study is to investigate molecular mechanisms of mitochondrial dysfunctions in hypertension-induced HFpEF as well as regulatory effects of PINK1 on mitochondrial fission. Here, we hypothesize that PINK1 phosphorylates Drp1^S616^ to improve mitochondrial fission, relieve mitochondrial dysfunctions, and slow the progression of hypertension-induced HFpEF. To test the hypothesis, Dahl salt-sensitivity (Dahl/SS) rats were fed with a high salt (HS) diet to develop hypertension-induced HFpEF. Physiological and biological experiments were carried out during the development of HFpEF. The significance and implications of the study were discussed for the treatment of HFpEF.

## 2. Results

### 2.1. Hypertension-Induced HFpEF Model

Figure 1A–D shows the significant increase of heart weight, left ventricular (LV) weight, and their ratio to body weight in the HF group. Figure 1F–I shows the increase of interventricular septal end diastolic thickness (IVSd) and E/E′ (the ratio of the maximum velocity of blood flow in the early diastolic phase of mitral valve to the motion velocity of mitral annulus) and the decrease of left ventricular end diastolic internal diameter (LVIDd), left ventricular end diastolic volume (EDV), and E/A (the ratio of early diastolic maximum velocity of mitral valve to systolic maximum velocity of atrium), denoting LV diastolic dysfunction in the HF group. There is no statistical difference of ejection fraction (EF) and fraction of shorten (FS) between control and HF groups. Figure 1L–O shows myocyte hypertrophy and aggravation of interstitial fibrosis in the HF group.

### 2.2. Mitochondrial Dysfunctions

To investigate the changes in mitochondria, Figure 2A shows the relevant genes, i.e., citrate synthase (CS), succeed dehydrogenase (SDH), pyruvate dehydrogenase (PDH), ATP synthase (ATPase), Sirtuin 3 (SIRT3), peroxisome promoter-activated receptor gamma coactivator-1 α (PGC-1α) and nuclear respiratory factor-1 (NRF-1) expressions. These genes are related to tricarboxylic acid cycle and maintain mitochondrial functions, which significantly decrease in HF group. There is also a decrease of ATP content, as shown in Figure 2B. These findings denote mitochondrial dysfunctions in hypertension-induced HFpEF rats. 

Figure 2C shows representative images of mitochondrial transmission electron microscopy (EM). The mitochondria change from a round shape to a longer/irregular shape in the HF group with decreased number and increased area (Figure 2D,E). Accordingly, Figure 2F,G shows a decrease in mitochondrial fission index and mRNA expression of Drp1 despite no significant changes in mRNA expression of Mfn2. The EM and mRNA data denote the decreased fission level and increased fusion level in mitochondria.

### 2.3. pDrp1S616-Mediated Mitochondrial Fission

To further determine the changes and mechanisms of mitochondrial fission, Figure 3A shows the decreased phosphorylation level of Drp1^S616^ in the HF group than the shams despite no statistical difference of total protein content of Drp1. After mitochondrial and cytoplasmic proteins were separated, Figure 3B shows the decreased level of Drp1 in mitochondria, but the increased level of Drp1 in cytoplasm in the HF group, which denotes the reduced mitochondrial localization of Drp1. Figure 3C,D show the reduced co-localization of Drp1 and mitochondria in myocytes of HFpEF rats. Hence, mitochondrial fission is reduced by hypertension-induced HFpEF.

### 2.4. PINK1-Regulated Mitochondrial Fission

The regulation of Drp1 by PINK1 (an upstream factor of Drp1) was investigated for understanding the changes of mitochondrial fission in hypertension-induced HFpEF. Figure 4A,B shows a significant decrease of PINK1 mRNA and protein levels in the LV of HF rats. Moreover, PINK1 was knocked out in H9C2 cardiomyocytes, which reduces the phosphorylation level of Drp1^S616^ significantly despite of no significant change in the total protein content of Drp1, as shown in Figure 4C,D. Figure 4E,F shows the decreased content of Drp1 in mitochondria and the increased content of Drp1 in cytoplasm, consistent with Figure 3. Figure 4G shows representative images of immunofluorescence of H9C2 cardiomyocytes. The co-localization degree of Drp1 and mitochondria decreases, as shown in Figure 4H. Representative images of MitoTracker Red fluorescent staining of mitochondrial morphology show the change of mitochondria from a round shape to a longer shape after PINK1 knockout in Figure 4I while the mitochondrial aspect ratio increases in Figure 4J. Hence, PINK1 knockout leads to the decrease of Drp1^S616^ phosphorylation-mediated mitochondrial fission. Figure 4K–N show the decreased expression of mitochondrial function related genes (CS, SDH, PDH, ATPase, and PGC-1α), mitochondrial membrane potential, and ATP content after PINK1 knockout, which denotes mitochondrial dysfunctions.

### 2.5. pDrp1S616-Mediated Mitochondrial Fission Regulates Mitochondrial Function

To confirm whether mitochondrial dysfunctions after PINK1 knockout is caused by the decrease of Drp1^S616^ phosphorylation-mediated mitochondrial fission, wild-type Drp1 (Drp1^WT^) and phosphorylated mutant Drp1 (Drp1^S616A^) were overexpressed on the basis of PINK1^KO^ cardiomyocytes. This significantly increased the total protein content of Drp1 while the overexpression of Drp1^WT^, rather than Drp1^S616A^, increased the phosphorylation level of Drp1^S616^ (Figure 5A). The overexpression of Drp1^WT^ also significantly increased the content of Drp1 on mitochondria and reversed the effect of PINK1 knockout despite of no changes owing to overexpression of Drp1^S616A^, as shown in Figure 5B,C. The overexpression of Drp1^WT^ increases the co-localization degree of Drp1 and mitochondria while overexpression of Drp1^S616A^ reduces it, as shown in Figure 5D,E. Hence, the mitochondrial localization of Drp1 is mainly caused by the phosphorylation of Drp1 S616 site. Accordingly, in Figure 5K,L, the change in mitochondrial morphometry shows that phosphorylation of Drp1 S616 site regulates mitochondrial fission. On the other hand, Figure 5F–J shows that overexpression of Drp1^WT^ but not overexpression of Drp1^S616A^, increases the expression of mitochondrial function-related genes (CS, SDH, PDH, ATPase, PGC-1α), mitochondrial membrane potential, and ATP content, and reverses the effect of PINK1 knockout, which shows that Drp1^S616^ phosphorylation can directly regulate mitochondrial function. These results indicate that Drp1^S616^ phosphorylation promotes mitochondrial localization of Drp1 and improves mitochondrial fission to relieve mitochondrial dysfunctions.

### 2.6. PINK1 Improves Hypertension-Induced HFpEF Phenotypes

To investigate effects of PINK1 on hypertension-induced HFpEF, PINK1 myocardium-specific overexpression was caused by adeno-associated virus (AAV) transfection to the HP group. Figure 6 shows that PINK1 overexpression can inhibit myocardial hypertrophy, aggravation of interstitial fibrosis and diastolic dysfunctions to slow the progression of HFpEF.

### 2.7. PINK1 Stimulates pDrp1S616-Mediated Mitochondrial Fission for Improvement of Mitochondrial Function

In comparison with the HF group, Figure 7A,B shows increased phosphorylation level of Drp1^S616^ and PINK1 mRNA expression as well as relative unchanged total protein content of Drp1; Figure 7C,D shows increased content of Drp1 in mitochondria and decreased content of Drp1 in cytoplasm; Figure 7E,F shows a higher co-localization degree of Drp1 and mitochondria in the HP group. Transmission electron microscope images also show increased number and decreased area of mitochondria (Figure 7G–J). Hence, PINK1 overexpression improves mitochondrial fission in HF rats. Moreover, Figure 7K,L shows increased expression of mitochondrial function-related genes (CS, SDH, PDH, ATPase, PGC-1α) and ATP content in the HP group. PINK1 overexpression activates Drp1^S616^ phosphorylation-mediated mitochondrial fission to improve the mitochondrial function.

## 3. Discussion

This study investigated molecular mechanisms of mitochondrial dysfunctions in hypertension-induced HFpEF relevant to the regulatory effect of PINK1. The findings were reported as: (1) hypertension-induced HFpEF impaired mitochondrial fission and led to mitochondrial dysfunctions; (2) In H9C2 cardiomyocytes, PINK1 knockout decreased phosphorylation level of Drp1^S616^ and mitochondrial localization of Drp1, inhibited mitochondrial fission, and induced mitochondrial dysfunctions; and (3) In HFpEF rats, myocardium-specific overexpression of PINK1 activated Drp1^S616^ phosphorylation, enhanced mitochondrial fission, and slowed the progression of HFpEF.

The hypertension-induced HFpEF model was established by feeding Dahl/SS rats a high salt diet, leading to risk factors such as hypertension, myocardial hypertrophy, diastolic dysfunctions, and interstitial fibrosis, consistent with previous studies [3,24,25,26]. Mitochondrial dysfunctions occurred during the occurrence and progression of HFpEF [27], which was caused by the imbalance between mitochondrial fusion and fission [28]. This study showed a decrease in expression of mitochondrial respiratory function-related genes, ATP production, and mitochondrial fission index as well as the altered mitochondrial morphometry, which characterized mitochondrial dysfunctions in hypertension-induced HFpEF rats. 

The phosphorylating Ser616 site of Drp1 is transferred to the mitochondrial outer membrane and bound to receptors, i.e., MFF and Fis1, to stimulate mitochondrial fissions [16]. We observed a decrease in phosphorylation level of Drp1^S616^ as well as mitochondrial localization of Drp1, which impaired mitochondrial fissions and induced mitochondrial dysfunctions in hypertension-induced HFpEF rats. Although elongated mitochondria with larger areas become more conducive to enhance mitochondrial function and promote ATP production [28], without fission, the over-fused mitochondria cannot remove damaged parts and hence induce HFpEF, consistent with studies of Shengnan Wu et al. [29] and Akihiro Shirakabe et al. [30]. 

PINK1, a serine/threonine protein kinase, is known to phosphorylate Drp1^S616^ [23]. In H9C2 cardiomyocytes, PINK1 knockout reduced the phosphorylation level of Drp1^S616^ and mitochondrial localization of Drp1. This resulted in a decrease in mitochondrial fission capacity and an increase in mitochondrial aspect ratio in agreement with the study of Hailong Han et al. [23]. On the other hand, PINK1 knockout decreased expression of mitochondrial function-related genes, mitochondrial membrane potential, and ATP production, showing mitochondrial dysfunctions. Moreover, in PINK1 knockout cells, overexpression of Drp1^WT^, but not overexpression of Drp1^S616A^, improved mitochondrial functions. The findings indicated the missing regulation of Drp1 on mitochondrial fission and mitochondrial function when the S616 site of Drp1 was not phosphorylated. Hence, PINK1 phosphorylates Drp1^S616^ to improve mitochondrial fission and restore mitochondrial function. Notably, PINK1 phosphorylates Mfn2 to inhibit mitochondrial fusion [31] and leads to mitochondrial elongation in HFpEF. The relevant mechanisms to both mitochondrial fusion and fission need to be confirmed in the following studies.

Myocardial-specific overexpression of PINK1 was achieved in hypertension-induced HFpEF rats by AAV. Phosphorylation level of Drp1^S616^ and mitochondrial localization of Drp1 increased after PINK1 overexpression. PINK1 overexpression also restored myocardial functions and reduced the degree of interstitial fibrosis. Hence, PINK1 myocardium-specific overexpression activated Drp1^S616^ phosphorylation and enhanced mitochondrial fission such that it slowed the progression of hypertension-induced HFpEF, as shown in Figure 8.

### Critiques of the Study

Since we showed mitochondrial dysfunctions in hypertension-induced HFpEF rats (*n* = 17 in HF vs. *n* = 10 in control) [24], we only selected four animals per group in the present study. The sample size is small, and should be increased in following studies. Although regulatory mechanism of PINK1/Drp1^S616^/mitochondrial pathway was showed in hypertension-induced HFpEF rats, the upstream of PINK1 is still unknown. Recent studies have shown that mitochondrial PINK1 expression in nucleus pulposus cells is significantly inhibited under high stress conditions [32]. In hypertension-induced HFpEF rats, heart and cardiomyocytes were continuously subjected to high stress. The high myocardial stress stimulated the decrease of PINK1 expression; however, an improved HFpEF animal model with the reasonable sample size is required to investigate regulatory mechanisms of PINK1 given that Dahl/SS rat is a hypertension-induced HFpEF model.

## 4. Materials and Methods

### 4.1. Animal Experiments

Twenty 6-week-old male Dahl/SS rats (Charles River, Beijing, China) were fed adaptively for one week. They were randomly divided into two groups. The heart failure group (HF) was fed with 8% NaCl high salt for 7 weeks, and the control group (**C**) was fed with 0.3% NaCl low salt for 7 weeks, with 4 rats in each group. The animals were kept in the Laboratory Animal Center of Shanghai Jiao Tong University and housed at 21–25 °C, with 40–50% humidity, a 12 h light: 12 h dark cycle, and free access to drinking water and eating standard feed. After feeding, ultrasonic testing was performed, and the body weight (BW) was weighed. Each rat was euthanized by cervical dislocation after inhalation anesthesia with isoflurane gas (concentration 5%, gas flow 500 mL/min). The rat heart was taken and the weight of the heart and LV were weighed after termination. All experiments were performed in accordance with Chinese National and Shanghai Jiao Tong University ethical guidelines regarding the use of animals in research, consistent with the NIH guidelines (Guide for the care and use of laboratory animals) on the protection of animals used for scientific purposes. The experimental protocol was approved by the Animal Care and Use Committee of Shanghai Jiao Tong University, China.

### 4.2. PINK1 Was Specifically Overexpressed by AAV In Vivo

The cDNA encoding PINK1 was cloned into the myocardial specific overexpression plasmid pLV [Exp]—cTnT (VectorBuilder, Guangzhou, China), and packaged with adeno-associated virus (AAV). The serotype was AAV9. After a week of adaptive feeding of 6-week-old male Dahl/SS rats, AAV was slowly injected into rats through the caudal vein, 1 × 10^12^ vg, followed by 7 weeks of high salt feeding. This resulted in PINK1 myocardial-specific overexpression heart failure rats (HP); Control rats (**C**) and heart failure rats (HF) were injected with empty AAV, with 4 rats in each group.

### 4.3. Echocardiography

Cardiac structure and function were determined using the VEVO 3100 high-resolution micro ultrasound system. After animals were anesthetized with isoflurane (concentration 5%, gas flow 500 mL/min), they were placed on the heating table in a supine position. Left ventricular end systolic internal diameter (LVIDs), LV end diastolic internal diameter (LVIDd), interventricular septal end systolic thickness (IVSs), interventricular septal end diastolic thickness (IVSd), LV end systolic volume (ESV), and LV end diastolic volume (EDV) were measured by the M-mode ultrasound, based on which ejection fraction (EF) and fraction of shorten (FS) were calculated; the ratio of early diastolic maximum velocity of mitral valve to systolic maximum velocity of atrium (E/A) was measured by the blood flow Doppler mode; the ratio of the maximum velocity of blood flow in the early diastolic phase of mitral valve to the motion velocity of mitral annulus (E/E′) was measured by the tissue Doppler mode.

### 4.4. Transmission Electron Microscope

The LV apical tissue was taken and fixed with 2.5% glutaraldehyde overnight. The heart was removed and post fixed in a mixture of 0.8% potassium ferrocyanide and 2% osmium tetroxide in 0.1 M sodium cacodylate buffer for 2 h. The area and number of mitochondria were observed by transmission electron microscope (Talo L120C G2). The decrease of mitochondrial number and the increase of mitochondrial area denoted the decrease of mitochondrial fission. The shooting times were 4300× and 13,500×. Each sample was photographed and analyzed with more than 5 visual fields and more than 100 mitochondria. The mitochondrial area was computed by ImageJ software.

### 4.5. H&E and Picro Sirius Red Staining

The isolated LV was fixed with 4% paraformaldehyde (China National Pharmaceutical Group Corporation, Beijing, China), embedded in paraffin, and sectioned to a thickness of 4 μm with a microtome (Leica RM2265, Wetztlar, Germany). After dewaxing, sections were stained with hematoxylin-eosin or Picro Sirius Red. Cell area and collagen fiber area were analyzed by ImageJ software.

### 4.6. Culture of H9C2 Cardiomyocyte Line and Lentivirus Transfection

H9C2 cardiomyocyte line was purchased from the National Collection of Authenticated Cell Cultures, China. The cells were cultured in DMEM medium (Gibco) supplemented with 10% fetal bovine serum and in an environment of 37 °C, 5% CO_2_, and 95% air.

PINK1^KO^ cells were generated by CRISPR/Cas9 system. PINK1 gRNA sequence: 5′-CCCGCACCACGAACTGCCGC-3′, which was cloned into knockout plasmid pLV [CRISPR] (VectorBuilder, China), and then packaged with lentivirus. In normal medium, lentivirus was transfected into H9C2 cardiomyocytes for 24 h to obtain PINK1^KO^ cells. Control cells (C) were transfected with empty lentivirus.

Production of Drp1^WT^ cells: the cDNA encoding Drp1 was cloned into the overexpression plasmid pLV [Exp] (VectorBuilder, China), and then packaged with lentivirus. In normal medium, lentivirus was transfected into PINK1^KO^ cells for 24 h to obtain Drp1^WT^ cells. Primer sequence of Drp1 cDNA: Drp1-F: 5′-CGCGGATCCGCCACCATGGAGGCGCTGATCC-3′; Drp1-R: 5′-CCGCTCGAGTCACCAAAGATGAGTCTCTCG-3′.

Production of Drp1^S616A^ cells: Drp1^S616A^ mutates TCT at Drp1 S616 site into GCT (alanine). The cDNA encoding Drp1^S616A^ was cloned into the overexpression plasmid pLV [Exp] (VectorBuilder, China), and then packaged with lentivirus. In normal medium, lentivirus was transfected into pink1ko cells for 24 h to obtain Drp1^S616A^ cells. The method of obtaining Drp1^S616A^ cDNA is as follows: using common cDNA as template, the first half of the target gene is obtained by using primers Drp1-1F: 5′-CGCGGATCCGCCACCATGGAGGCGCTGATCC-3′, Drp1-1R: 5′-GGCAGCCAGTTTTCGTG-3′. The second half of the target gene was obtained by primers Drp1-2F: 5′-CTGGCTGCCCGAGAAC-3′, Drp1-2R: 5′-CCGCTCGAGTCACCAAAGA-TGAGTCTCTCG-3′. Using the above product as the template, the cDNA of Drp1^S616A^ was obtained by using primers Drp1-F: 5′-CGCGGATCCGCCACCATGGAGGCGCTGATCC-3′, Drp1-R: 5′-CCGCTCGAGTCACCAAAGATGAGTCTCTCG-3′.

### 4.7. Real-Time PCR

The LV was weighed (40 mg), cut, and placed in a 2 mL centrifuge tube, and 1 mL Trizol (Invitrogen) was added to extract RNA (if the sample was 6-well plate cells, 500 ml Trizol were used to extract RNA). cDNA synthesis was performed according to the instruction of the RevertAid™ First-Strand cDNA Synthesis Kit. The primers were synthesized by Sangon Biotech Co., Ltd. (Shanghai, China). (Appendix A). Amplification was performed using a real-time PCR instrument (ABI StepOnePlus Real Time PCR System 7500, Los Angeles, CA, USA). The reaction conditions were as follows: 95 °C for 10 min, then 40 cycles of denaturation at 95 °C for 15 s and 60 °C for 1 min. 

### 4.8. Separation of Mitochondrial Protein and Cytoplasmic Protein

Take 50 mg LV tissue, and separate mitochondrial protein and cytoplasmic protein according to the process of mitochondrial separation Kit (Beyotime Biotechnology, Shanghai, China). After the sample was washed with the precooled PBS, mitochondrial separation reagent was added and homogenized; 1000 g of supernatant was centrifuged for 5 min to remove the precipitation, 11,000 g of supernatant was centrifuged for 10 min to obtain mitochondrial precipitation, and the remaining supernatant was cytoplasmic protein.

### 4.9. Western Blot

LV tissue (20 mg) was lysed with Ripa lysate (Beyotime Biotechnology) and the total protein was extracted. SDS-PAGE electrophoresis and membrane transfer were performed. Primary antibodies: Drp1 (1:1000, Abcam, Cambridge, UK), Drp1^S616^ (1:1000, CST, Boston, MA, USA), PINK1 (1:1000, CST, Boston, MA, USA), β-actin (1:1000, protein, USA), voltage-dependent anion channel protein 1 (VDAC1, 1:1000, Immunoway, Suzhou, China). The strips were exposed with Fusion FX automatic chemiluminescence/fluorescence image analysis system, and the gray value of the strips was measured by ImageJ analysis software.

### 4.10. Immunofluorescence

The paraffin section was dewaxed and heated in a 95 °C water bath for 20 min in sodium citrate antigen repair solution (Beyotime Biotechnology) and naturally cooled to room temperature (cell samples were fixed with 4% paraformaldehyde for 15 min). Sections or cells were blocked for 2 h at room temperature and incubated at 4 °C for 20 h with primary antibodies against Drp1 (1:200, Abcam, UK) and mitochondrial preprotein translocases of the outer membrane (Tom20, 1:100, Santa Cruz, CA, USA). After washing, they were incubated with Alexa Fluor 568 (1:1000, Jackson, PA, USA) and Alexa Fluor 488 (1:1000, Jackson, PA, USA) secondary antibodies for 1 h at room temperature in the dark. After washing, they were incubated with DAPI (Beyotime Biotechnology) for 5 min. Sections or cells were imaged with a confocal microscope (Olympus Fluoview FV1000, Tokyo, Japan). More than 5 pictures were taken for each sample. Cell samples were taken with more than 30 cells in each group. In each image, the ImageJ plug-in Coloc 2 was used to detect the Manders co-localization Coefficient (MCC) and to evaluate the co-localization of mitochondria and Drp1.

### 4.11. Morphological Analysis of Mitochondria

The cells were inoculated into the culture dish for overnight culture and washed twice with PBS. The culture containing 100 nM MitoTracker Red (Beyotime Biotechnology) was added. The cells were incubated at 37 °C for 20 min and washed with PBS 3 times. After replacing the culture medium, fluorescence images were obtained with the confocal microscope (Olympus Fluoview FV1000). More than 30 cells were photographed in each group. The mitochondrial aspect ratio was analyzed by ImageJ software to evaluate mitochondrial fission. The increase of aspect ratio indicated the prolongation of mitochondria, which is a sign of the decrease of mitochondrial fission.

### 4.12. Mitochondrial Membrane Potential

Mitochondrial membrane potential was detected by tetramethylrhodamine ethyl ester (TMRE) kit (Beyotime Biotechnology). The cells were inoculated into the Petri dish for overnight culture, washed twice with PBS, and incubated at 37 °C with TMRE working solution for 15 min. The preheated medium was washed twice. After replacing the new medium, the fluorescence image was taken with the confocal microscope. More than 30 cells were photographed in each group. The fluorescence intensity of mitochondrial membrane potential was analyzed by ImageJ software v2.3.0.

### 4.13. ATP Content Detection

A total of 20 mg of LV tissue or an equal amount of 6-well plate cells were taken to detect the ATP content, according to the process outlined in the ATP detection kit (Beyotime Biotechnology). The sample was homogenized after adding 200 ul lysate and centrifuged at 12,000× *g* for 5 min to obtain ATP. After adding ATP detection reagent, the sample luminous intensity was analyzed by chemiluminescence instrument to detect the ATP content. The protein concentration was detected. The ATP content of each sample was standardized through the protein concentration to obtain the relative content of ATP.

### 4.14. Statistical Analysis

The two groups of data were analyzed by the independent sample *t*-test. One-way ANOVA test was used for the data of 3 groups and above, and Bonferroni or Dunnett’s T3 was used for pairwise comparison. *p* < 0.05 indicated the significant difference. All data were analyzed using SPSS Statistics V21.0 software and the results were expressed as the mean ± standard deviation (SD).

## 5. Conclusions

We carried out physiological and biological experiments in hypertension-induced HFpEF rats. HFpEF caused mitochondrial dysfunctions relevant to the impairment of mitochondrial fission. PINK1 knockout in H9C2 cardiomyocytes also impaired mitochondrial fission and functions. Furthermore, PINK1 myocardium-specific overexpression to animals was found to activate Drp1^S616^ phosphorylation and enhance mitochondrial fission, which slowed the progression of hypertension-induced HFpEF. Hence, PINK1 can phosphorylate Drp1^S616^ to improve mitochondrial fission, relieve mitochondrial dysfunctions, and slow the progression of hypertension-induced HFpEF.

## Figures and Tables

**Figure 1 ijms-23-11934-f001:**
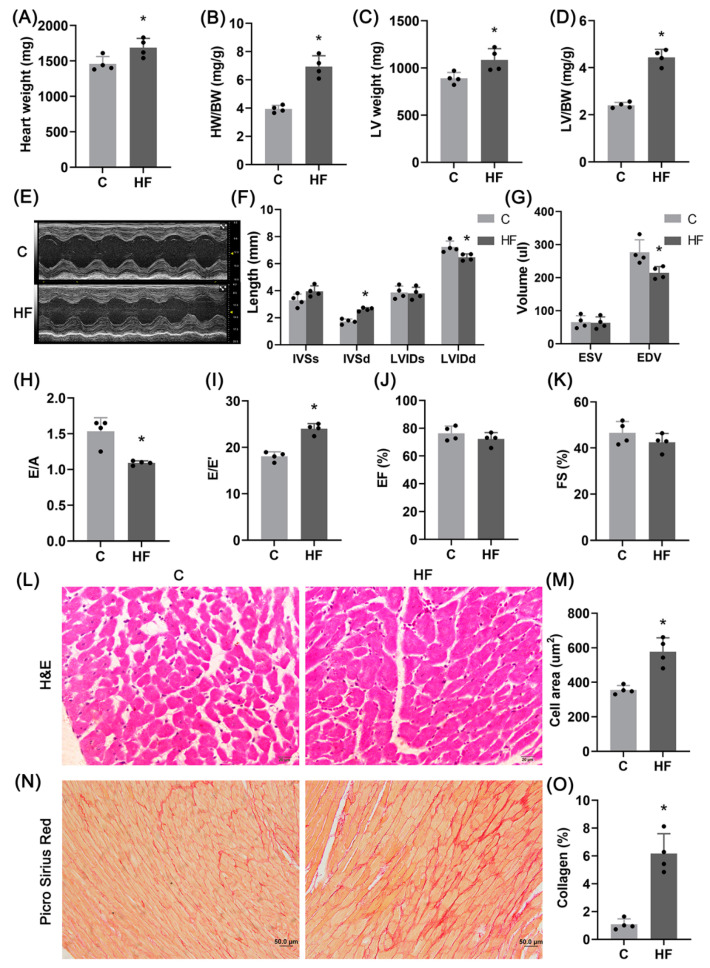
Hypertension-induced HFpEF model. (**A**–**D**) Heart weight (**A**), ratio of HW to BW (**B**), LV weight (**C**), and ratio of LV weight to BW (**D**); *n* = 4 per group. (**E**–**K**) Representative images of M-mode ultrasound (**E**), interventricular septal end systolic thickness (IVSs), interventricular septal end diastolic thickness (IVSd), LV end systolic internal diameter (LVIDs), LV end diastolic internal diameter (LVIDd) (**F**), end-systolic volume (ESV), end-diastolic volume (EDV) (**G**), E/A (**H**), E/E′ (**I**), ejection fraction (EF) (**J**), and fraction of shortened (FS) (**K**); *n* = 4 per group. (**L**,**M**) Representative images of LV H&E staining (**L**) and quantitative analysis of myocardial cell area (4 samples per group, 3–5 fields of view per sample, Scale bar: 20 μm) (**M**). (**N**,**O**) Representative images of Picro Sirius Red staining (**N**) and quantitative analysis of collagen fiber area fraction (4 samples per group, 3–5 fields analyzed per sample, Scale bar: 50 μm) (**O**). The data are mean ± SD, * Compared with the C group, *p* < 0.05.

**Figure 2 ijms-23-11934-f002:**
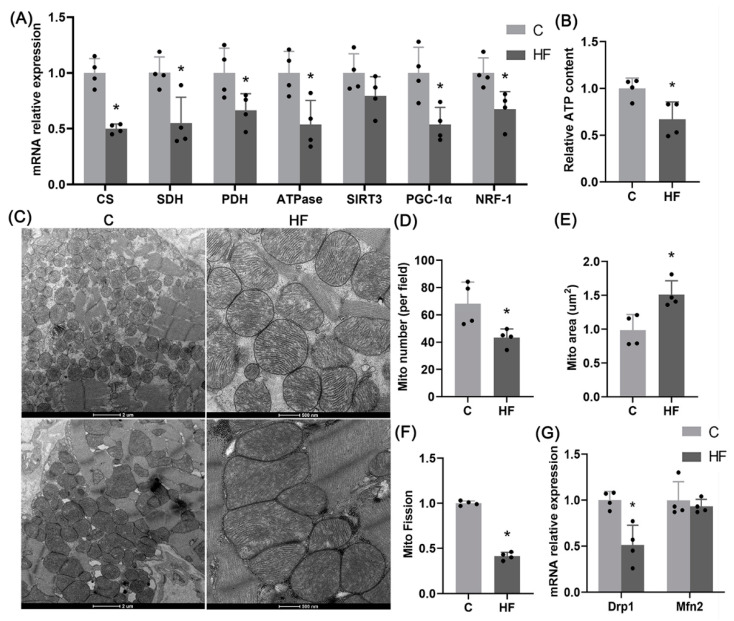
Mitochondrial dysfunctions in hypertension-induced HFpEF. (**A**) mRNA expression of mitochondrial function-related genes CS, SDH, PDH, ATPase, SIRT3, PGC-1α, NRF-1 in the LV; *n* = 4 per group. (**B**) Relative ATP content in the LV (normalized using protein concentration); *n* = 4 per group. (**C**–**F**) Representative images of mitochondrial transmission electron microscopy in the LV (Scale bar: 2 μm, 500 nm) (**C**), quantitative analysis of the number of mitochondria per unit field of view (analysis of more than 100 mitochondria per sample, 4 samples per group) (**D**), quantitative analysis of mitochondrial area (analysis of more than 100 mitochondria per sample, 4 samples per group) (**E**), and quantitative analysis of mitochondrial fission index (ratio of mitochondrial number to mitochondrial area, *n* = 4 per group) (**F**). (**G**) mRNA expression of Drp1, a key gene for mitochondrial fission in the left ventricle of rats, and Mfn2, a key gene for mitochondrial fusion (*n* = 4 in each group). The data are mean ± SD, * Compared with the C group, *p* < 0.05.

**Figure 3 ijms-23-11934-f003:**
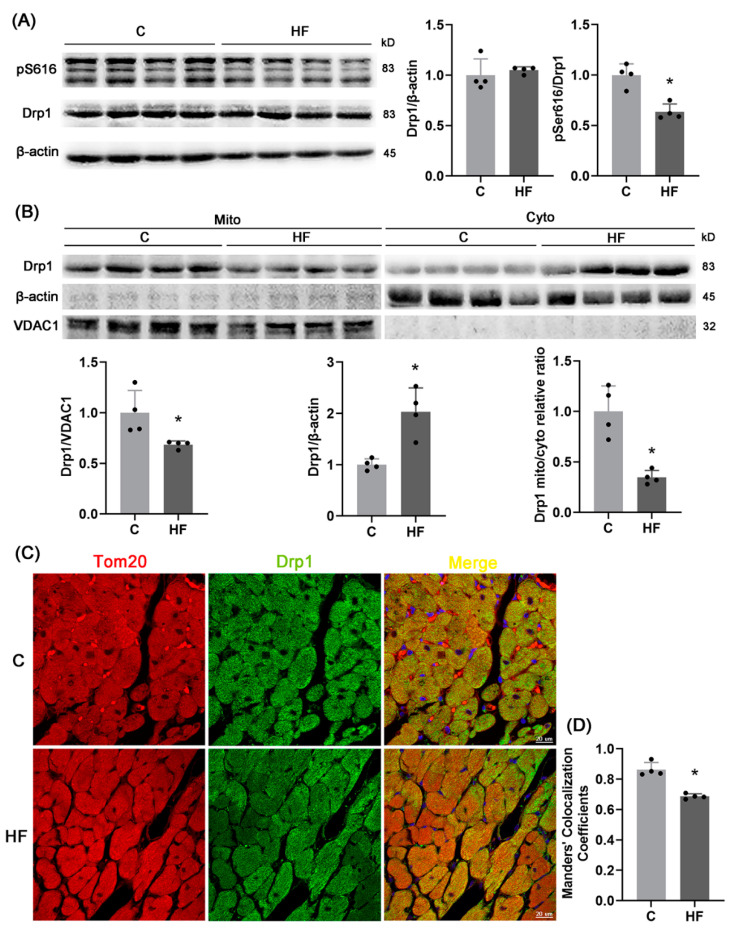
pDrp1^S616^-mediated mitochondrial fission. (**A**) WB analysis of total protein levels of Drp1 and phosphorylation of Drp1^S616^ in the LV; *n* = 4 per group. (**B**) Drp1 protein level on mitochondria, Drp1 protein level in cytoplasm, and relative ratio of mitochondrial Drp1 to cytoplasmic Drp1 after the separation of LV mitochondria and cytoplasmic proteins; *n* = 4 per group. (**C**) Representative images of immunofluorescence of the LV (red: mitochondrial outer membrane marker, Tom20, green: Drp1, blue: DAPI, Scale bar: 20 μm), and (**D**) Manders co-localization coefficient (MCC) quantitative analysis of Tom20 and Drp1 (5 fields analyzed per sample, 4 samples per group). The data are mean ± SD, * Compared with the C group, *p* < 0.05.

**Figure 4 ijms-23-11934-f004:**
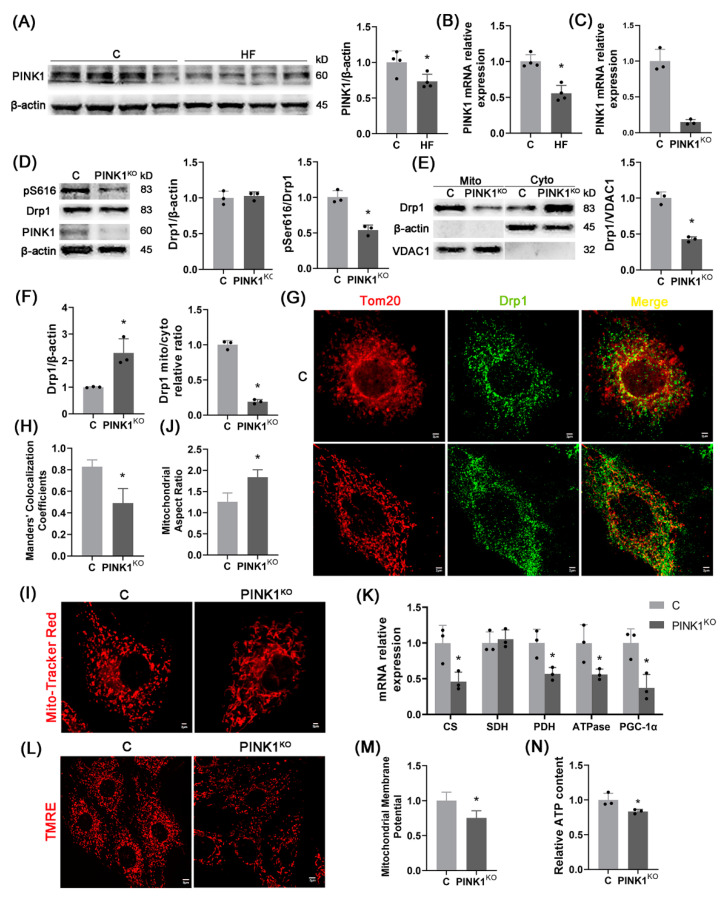
PINK1 knockout in H9C2 cardiomyocytes impairs mitochondrial fission and function. (**A**,**B**) PINK1 protein expression (**A**) and mRNA expression (**B**) in the LV; *n* = 4 per group. (**C**,**D**) PINK1 mRNA expression (**C**) and PINK1 protein expression, Drp1 total protein level and Drp1^S616^ phosphorylation level (**D**) in H9C2 cardiomyocytes after PINK1 knockout; *n* = 3 per group. (**E**,**F**) Drp1 protein level on mitochondria (**E**) and Drp1 protein level in cytoplasm and relative ratio of mitochondrial Drp1 to cytoplasmic Drp1 (**F**) after separation of mitochondrial and cytoplasmic proteins of H9C2 cardiomyocytes; *n* = 3 per group. (**G**,**H**) Representative images of immunofluorescence of H9C2 cardiomyocytes (red: mitochondrial outer membrane marker, Tom20, green: Drp1, blue: DAPI, Scale bar: 2 μm) (**G**) and Manders co-localization coefficient (MCC) quantitative analysis of Tom20 and Drp1 (more than 30 cells were analyzed per group) (**H**). (**I**,**J**) Representative images of MitoTracker Red fluorescent staining of mitochondrial morphology (Scale bar: 2 μm) (**I**) and quantitative analysis of mitochondrial aspect ratio (more than 30 cells analyzed per group) (**J**). (**K**) mRNA expression of mitochondrial function-related genes CS, SDH, PDH, ATPase, and PGC-1α in H9C2 cardiomyocytes, *n* = 3 per group. (**L**,**M**) Representative images of TMRE mitochondrial membrane potential fluorescence staining (Scale bar: 5 μm) (**L**) and quantitative analysis of mitochondrial membrane potential fluorescence intensity (more than 30 cells analyzed per group) (**M**). (**N**) Relative ATP content of H9C2 cardiomyocytes (normalized using protein concentration, *n* = 3 per group). The data are mean ± SD, * Compared with the C group, *p* < 0.05.

**Figure 5 ijms-23-11934-f005:**
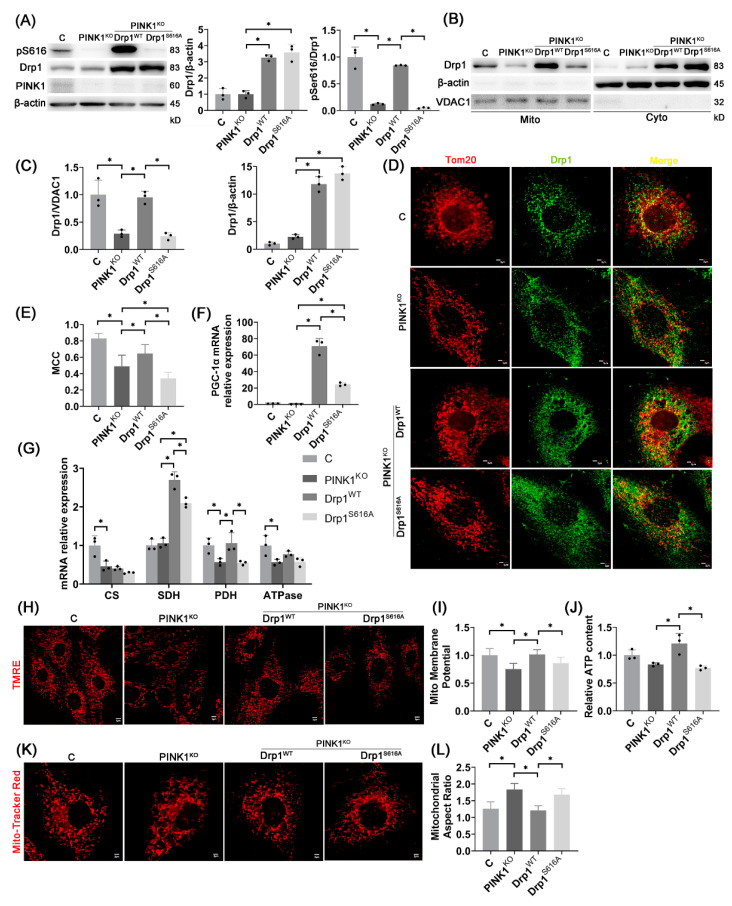
Overexpression of Drp1^WT^, rather than Drp1^S616A^, improve mitochondrial fission and function. (**A**) PINK1 protein expression, Drp1 total protein level and Drp1^S616^ phosphorylation level after PINK1^KO^ cardiomyocytes overexpressed Drp1^WT^ or Drp1^S616A^; *n* = 3 per group. (**B**,**C**) Drp1 protein levels on mitochondria and in cytoplasm after separation of mitochondrial and cytoplasmic proteins of H9C2 cardiomyocytes; *n* = 3 per group. (**D**,**E**) Representative images of immunofluorescence of H9C2 cardiomyocytes (red: mitochondrial outer membrane marker, Tom20, green: Drp1, blue: DAPI, Scale bar: 2 μm) (**D**) and Manders co-localization coefficient (MCC) quantitative analysis of Tom20 and Drp1 (more than 30 cells analyzed per group) (**E**). (**F**,**G**) mRNA expression of mitochondrial function-related gene PGC-1α (**F**) and CS, SDH, PDH, ATPase (**G**) in H9C2 cardiomyocytes, *n* = 3 per group. (**H**,**I**) Representative images of TMRE mitochondrial membrane potential fluorescence staining (Scale bar: 5 μm) (**H**) and quantitative analysis of mitochondrial membrane potential fluorescence intensity (more than 30 cells were analyzed per group) (**I**). (**J**) Relative ATP content of H9C2 cardiomyocytes (normalized using protein concentration, *n* = 3 per group). (**K**,**L**) Representative images of MitoTracker Red fluorescent staining of mitochondrial morphology (Scale bar: 2 μm) (**K**) and quantitative analysis of mitochondrial aspect ratio (more than 30 cells analyzed per group) (**L**). The data are mean ± SD, * *p* < 0.05 compared to the other group.

**Figure 6 ijms-23-11934-f006:**
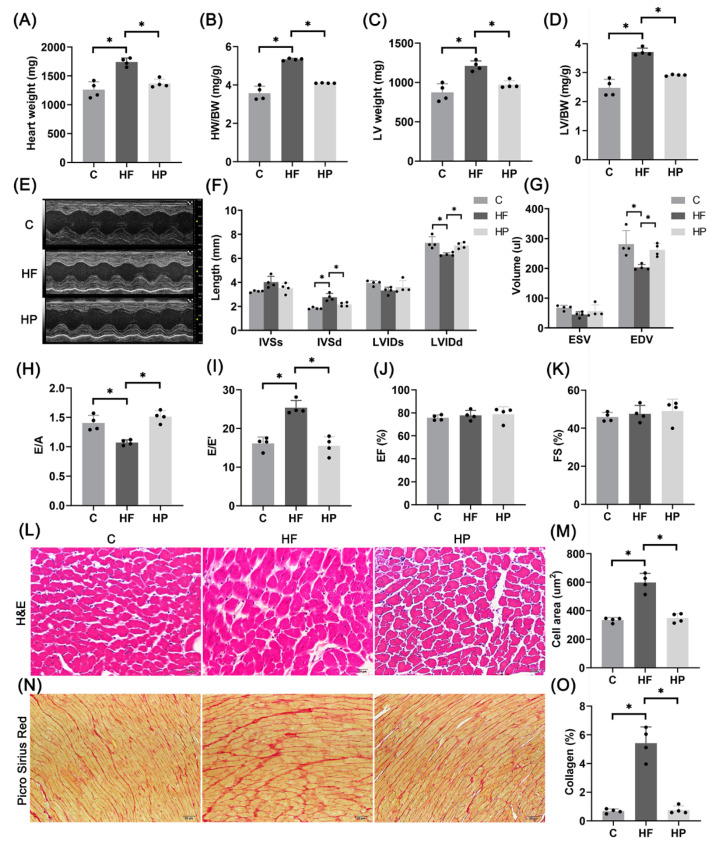
PINK1 overexpression improves hypertension-induced HFpEF phenotypes. (**A**–**D**) Heart weight (**A**), ratio of heart weight to body weight (**B**), left ventricle weight (**C**), and ratio of left ventricle weight to body weight (**D**); *n* = 4 per group. (**E**–**K**) Representative images of M-mode ultrasound (**E**), IVSs, IVSd, LVIDs, LVIDd (**F**), ESV, EDV (**G**), E/A (**H**), E/E′ (**I**), EF (**J**), and FS (**K**); *n* = 4 per group. (**L**,**M**) Representative images of H&E staining of rat left ventricle (**L**) and quantitative analysis of myocardial cell area (4 samples per group, 3–5 fields of view per sample, Scale bar: 20 μm) (**M**). (**N**,**O**) Representative images of Picro Sirius Red staining (**N**) and quantitative analysis of collagen fiber area fraction (4 samples per group, 3–5 fields analyzed per sample, Scale bar: 50 μm) (**O**). The data are mean ± SD, * *p* < 0.05 compared to the other group.

**Figure 7 ijms-23-11934-f007:**
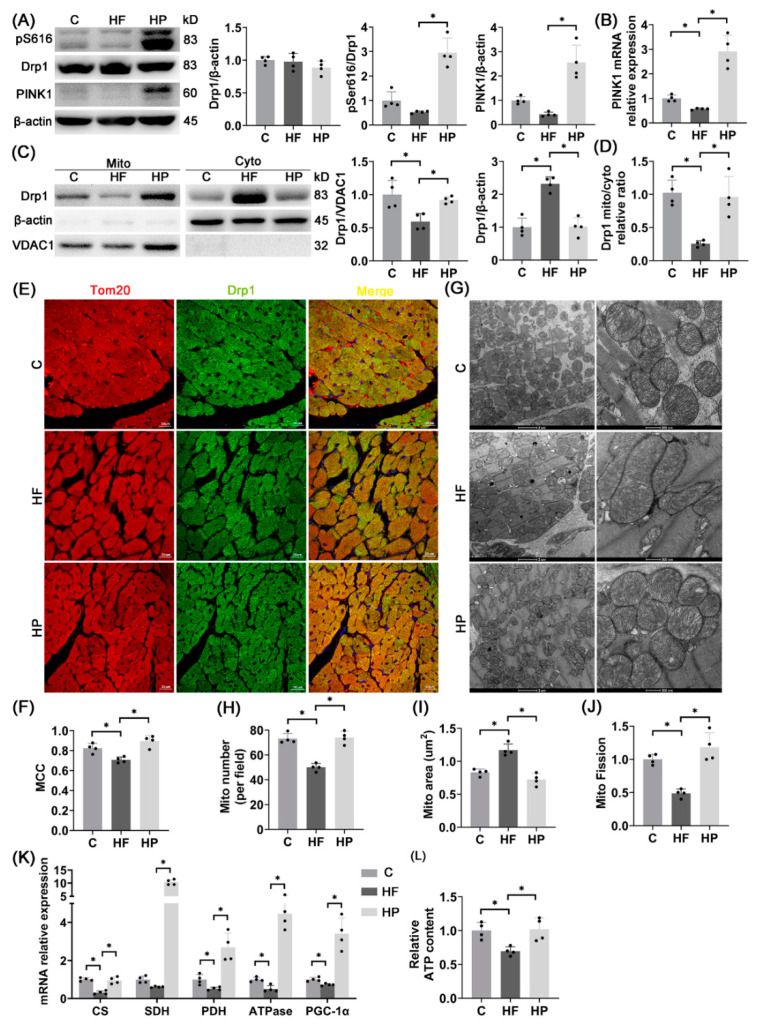
PINK1 overexpression stimulates pDrp1^S616^-mediated fission to improve mitochondrial function. (**A**) Drp1 total protein expression, Drp1^S616^ phosphorylation level, and PINK1 protein expression in the LV; *n* = 4 per group. (**B**) PINK1 mRNA expression in the LV; *n* = 4 per group. (**C**,**D**) Drp1 protein level on mitochondria, Drp1 protein level in cytoplasm (**C**) and the relative ratio of mitochondrial Drp1 to cytoplasmic Drp1 (**D**) after the separation of LV mitochondria and cytoplasmic proteins; *n* = 4 per set. (**E**,**F**) Representative images of immunofluorescence of the LV (red: mitochondrial outer membrane marker, Tom20, green: Drp1, blue: DAPI, Scale bar: 20 μm) (**E**) and Manders co-localization coefficient (MCC) quantitative analysis of Tom20 and Drp1 (5 fields analyzed per sample, 4 samples per group) (**F**). (**G**–**J**) Representative images of mitochondrial transmission electron microscopy in the LV (Scale bar: 2 μm, 500 nm) (**G**) and quantitative analysis of the number of mitochondria per unit field of view (analysis of >100 mitochondria per sample, 4 samples per group) (**H**), quantitative analysis of mitochondrial area (more than 100 mitochondria analyzed per sample, 4 samples per group) (**I**), and quantitative analysis of mitochondrial fission index (ratio of mitochondrial number to mitochondrial area, *n* = 4 per group) (**J**). (**K**) mRNA expression of mitochondrial function-related genes CS, SDH, PDH, ATPase and PGC-1α in the LV; *n* = 4 per group. (**L**) Relative ATP levels in rat hearts (normalized using protein concentration); *n* = 4 per group. The data are mean ± SD, * *p* < 0.05 compared to the other group.

**Figure 8 ijms-23-11934-f008:**
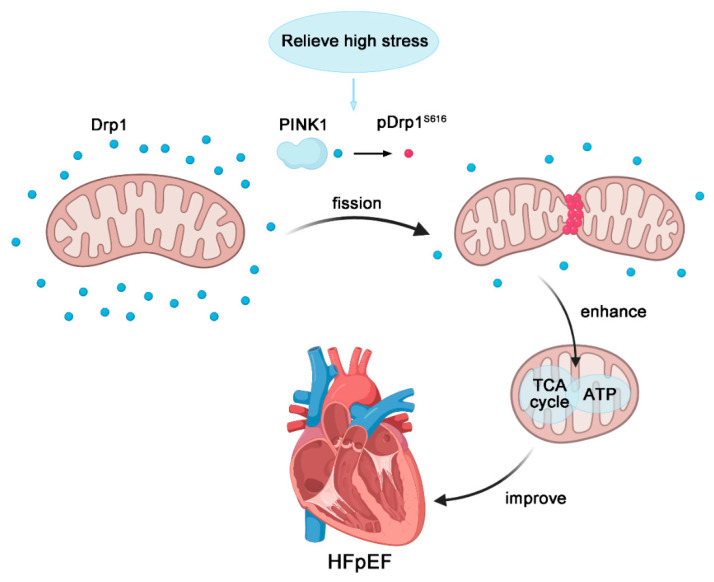
Molecular mechanism of PINK1-regulated hypertension-induced HFpEF. PINK1 increases phosphorylation level of Drp1^S616^ and mitochondrial localization of Drp1 to stimulate mitochondrial fission, thereby restoring mitochondrial function and ultimately slowing the development of hypertension-induced HFpEF.

## Data Availability

All data generated or analyzed during this study are included in this published article (Appendix A at DOI: https://doi.org/10.6084/m9.figshare.21005899 and https://doi.org/10.6084/m9.figshare.21005752 (accessed on 9 July 2022).

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
