# Peer review of "PINK1 Phosphorylates Drp1S616 to Improve Mitochondrial Fission and Inhibit the Progression of Hypertension-Induced HFpEF"

_ijms, 2022, doi:10.3390/ijms231911934_

Round 1

Reviewer 1 Report

The manuscript „PINK1 Phosphorylates Drp1S616 to Improve Mitochondrial Fission and Inhibit the Progression of hypertension-induced HFpEF” by Jian Shou and Yunlong Huo discusses the role of PINK1 in mitochondrial induced HFpEF to possibly find specific therapeutical measures in this disease entity. There are a few points to be considered. The manuscript has been written very well. The experiments have been designed and performed well. 

Please write out abbreviations when first mentioned in the manuscript (e.g. HS, line 51).

Author Response

Point 1: Please write out abbreviations when first mentioned in the manuscript (e.g. HS, line 51).

Response: We revised the text as suggested. Please see the below revisions. Thank you.

Changes to Ms. (Lines 57-65)

“Figures 1A-D show the significant increase of heart weight, left ventricular (LV) weight and their ratio to body weight in the HF group. Figures 1F-I show the increase of interventricular septal end diastolic thickness (IVSd) and E/E' (the ratio of the maximum velocity of blood flow in the early diastolic phase of mitral valve to the motion velocity of mitral annulus) and the decrease of left ventricular end diastolic internal diameter (LVIDd), left ventricular end diastolic volume (EDV) and E/A (the ratio of early diastolic maximum velocity of mitral valve to systolic maximum velocity of atrium), denoting LV diastolic dysfunction in the HF group. There is no statistical difference of ejection fraction (EF) and fraction of shorten (FS) between control and HF groups.”

Changes to Ms. (Lines 202-204)

“To investigate effects of PINK1 on hypertension-induced HFpEF, PINK1 myocardium-specific overexpression was caused by adeno-associated virus (AAV) transfection to the HP group.”

Reviewer 2 Report

Minor concerns:

- authors need to increase animal group studies , n=4 it's not enough to determinate the phenotype 

- paragraph 2.2 authors need to explain better EM and mRNA data, moreover they need to explain in 2-3 sentences related to PINK rules on mitofusin2 there is a recent evidence in "https://doi.org/10.3389/fcell.2022.868465"  

-Figure 8 can be improved to make molecular mechanism more immediate to understand 

Author Response

Point 1: authors need to increase animal group studies, n=4 it's not enough to determinate the phenotype.

Response: We agree with the reviewer that the sample size is small. Since we previously showed mitochondrial dysfunctions in hypertension-induced HFpEF rats (n=17 in HF vs n=10 in control) [Int J Mol Sci 2020, 21, 3362, doi:10.3390/ijms21093362], we only selected 4 animals per group in the present study. We will use an improved HFpEF animal model with the reasonable sample size in the following studies. Please see the below revisions. Thank you.

Changes to Ms. (Lines 301-303)

“Since we showed mitochondrial dysfunctions in hypertension-induced HFpEF rats (n=17 in HF vs n=10 in control) [24], we only selected 4 animals per group in the present study. The sample size is small, which should increase in the following studies.”

Changes to Ms. (Lines 308-311)

“The high myocardial stress stimulated the decrease of PINK1 expression, an improved HFpEF animal model with the reasonable sample size is required to investigate regulatory mechanisms of PINK1 given that Dahl/SS rat is a hypertension-induced HFpEF model.”

Point 2: paragraph 2.2 authors need to explain better EM and mRNA data, moreover they need to explain in 2-3 sentences related to PINK rules on mitofusin2 there is a recent evidence in "https://doi.org/10.3389/fcell.2022.868465".

Response: We explain EM and mRNA data in details in paragraph 2.2 and add sentences in the section ‘Discussion’ as suggested. Please see the below revisions. Thank you.

Changes to Ms. (Lines 80-95)

“To investigate the changes of mitochondria, Figure 2A shows the relevant genes, i.e., citrate synthase (CS), succeed dehydrogenase (SDH), pyruvate dehydrogenase (PDH), ATP synthase (ATPase), Sirtuin 3 (SIRT3), peroxisome promoter activated receptor gamma coactivator-1 α (PGC-1α) and nuclear respiratory factor-1 (NRF-1) expressions. These genes are related to tricarboxylic acid cycle and maintain mitochondrial functions, which significantly decrease in HF group. There is also a decrease of ATP content, as shown in Fig. 2B. These findings denote mitochondrial dysfunctions in hypertension-induced HFpEF rats.

Figure 2C shows representative images of mitochondrial transmission electron microscopy (EM). The mitochondria change from round shape to longer/irregular shape in the HF group with decreased number and increased area (Figs. 2D and E). Accordingly, Figures 2F and G show a decrease of mitochondrial fission index and mRNA expression of Drp1 despite of no significant changes in mRNA expression of Mfn2. The EM and mRNA data denote the decreased fission level and increased fusion level in mitochondria.”

Changes to Ms. (Lines 286-288)

“Notably, PINK1 phosphorylates Mfn2 to inhibit mitochondrial fusion [31] and leads to mitochondrial elongation in HFpEF. The relevant mechanisms to both mitochondrial fusion and fission need to be confirmed in the following studies.”

Point 3: Figure 8 can be improved to make molecular mechanism more immediate to understand.

Response: We revised Fig. 8 as suggested. Please see the revised figure in the manuscript.